# Risk Factors and Corresponding Management for Suture Anchor Pullout during Arthroscopic Rotator Cuff Repair

**DOI:** 10.3390/jcm11226870

**Published:** 2022-11-21

**Authors:** Xiangwei Li, Yujia Xiao, Han Shu, Xianding Sun, Mao Nie

**Affiliations:** Center for Joint Surgery, Department of Orthopaedics, The Second Affiliated Hospital of Chongqing Medical University, Chongqing 400010, China

**Keywords:** rotator cuff tear, rotator cuff repair, bone quality, osteopenia, osteoporosis, anchor pullout, pullout strength

## Abstract

Introduction: Due to the aging of the population, the incidence of rotator cuff tears is growing. For rotator cuff repair, arthroscopic suture-anchor repair has gradually replaced open transosseous repair, so suture anchors are now considered increasingly important in rotator cuff tear reconstruction. There are some but limited studies of suture anchor pullout after arthroscopic rotator cuff repair. However, there is no body of knowledge in this area, which makes it difficult for clinicians to predict the risk of anchor pullout comprehensively and manage it accordingly. Methods: The literature search included rotator cuff repair as well as anchor pullout strength. A review of the literature was performed including all articles published in PubMed until September 2021. Articles of all in vitro biomechanical and clinical trial levels in English were included. After assessing all abstracts (*n* = 275), the full text and the bibliographies of the relevant articles were analyzed for the questions posed (*n* = 80). Articles including outcomes without the area of interest were excluded (*n* = 22). The final literature research revealed 58 relevant articles. Narrative synthesis was undertaken to bring together the findings from studies included in this review. Result: Based on the presented studies, the overall incidence of anchor pullout is not low, and the incidence of intraoperative anchor pullout is slightly higher than in the early postoperative period. The risk factors for anchor pullout are mainly related to bone quality, insertion depth, insertion angle, size of rotator cuff tear, preoperative corticosteroid injections, anchor design, the materials used to produce anchors, etc. In response to the above issues, we have introduced and evaluated management techniques. They include changing the implant site of anchors, cement augmentation for suture anchors, increasing the number of suture limbs, using all-suture anchors, using an arthroscopic transosseous knotless anchor, the Buddy anchor technique, Steinmann pin anchoring, and transosseous suture repair technology. Discussion: However, not many of the management techniques have been widely used in clinical practice. Most of them come from in vitro biomechanical studies, so in vivo randomized controlled trials with larger sample sizes are needed to see if they can help patients in the long run.

## 1. Introduction

Due to the aging of the population, the incidence of rotator cuff tears is growing [1,2]. For rotator cuff repair, arthroscopic suture-anchor repair has gradually replaced open transosseous repair, so suture anchors are now considered increasingly important in rotator cuff tear reconstruction [3]. The majority of patients with rotator cuff tears are over 60 years old, and osteoporosis is very common among them [4,5]. This means that their proximal humeral bone quality is often poor, which will increase the incidence of anchor pullout [6,7] (Figure 1). Anchor pullout is one of the mechanisms of suture anchor failure. It occurs at the anchor-bone interface during arthroscopic rotator cuff repair, resulting in pullout of the anchors from the bone [8]. In terms of biomechanics, pullout strength is the pullout force measured when anchor pullout occurs at the anchor-bone interface. Studies have attempted to find new methods to improve pullout strength, thus reducing the risk of anchor pullout.

The purpose of this review is to summarize the body of knowledge on suture anchor pullout during arthroscopic rotator cuff repair. We will first briefly introduce the incidence of anchor pullout before discussing the reason why the suture anchor pulls out. Lastly, we will describe in detail the different technologies and studies that have been used to solve the problem of anchor pullout, and we will compare their pros and cons to help future practice.

## 2. Method

The literature search included rotator cuff repair as well as anchor pullout strength. A review of the literature was performed, including all articles published in PubMed until September 2021. The following search terms were used alone and in combination: “Rotator cuff repair”, “anchor pullout”, and “pullout strength”. The articles were assessed considering the following research aspects: definition, incidence of anchor pullout, risk factor for anchor pullout, and anchor pullout management.

Articles of all in vitro biomechanical and clinical trial levels in English were included. After assessing all abstracts (*n* = 275), the full text and the bibliographies of the relevant articles were analyzed for the questions posed (*n* = 80). Articles including outcomes without the area of interest were excluded (*n* = 22). Specifically, publications from 1990 to 2021 were included because of the advancements in biomechanics, surgical treatments, and improved understanding of pullout strength for suture anchors. The final literature research revealed 58 relevant articles. Narrative synthesis was undertaken to bring together the findings from studies included in this review.

## 3. Results

### 3.1. Total Incidence of Anchor Pullout

The incidence of anchor pullout varies depending on the circumstances. The incidence of early anchor pullout after arthroscopic rotator cuff repair is approximately 0.1%–3.1%, while the incidence of anchor pullout during surgery is higher, approximately 3.3%–5.4%. Anchor pullout is one of the three mechanical mechanisms of revision surgery failure, with an incidence of approximately 4.5%. 

#### 3.1.1. Early Anchor Pullout

A retrospective, monocentric study [9] by Skaliczki et al., showed that early anchor pullout was observed in six patients out of 5327 (0.1%).

Earlier studies found much higher rates of early anchor pullout. Benson et al. [10] investigated 269 patients who underwent arthroscopic rotator cuff repair and found six cases of early anchor pullout (2.4%). In their study of 127 patients, Dezaly et al. [11] reported a 3.1% prevalence of early anchor displacement. The difference may be caused by the different time points of the radiographic evaluation: The time period considered in the latter two studies also included part of the rehabilitation, while the observation time of Skaliczki et al. [9] was immediately after surgery. That is, early anchor pullout in the series of Skaliczki et al. [9] is mainly attributed to surgical intervention, while the results of Benson et al. [10] and Dezaly et al. [11] are at least partly due to the rehabilitation process.

#### 3.1.2. Anchor Pullout during Surgery

The incidence of anchor pullout during surgery has not been investigated, and only a few articles [9,10,11] have reported the incidence of early postoperative anchor failure.

A retrospective study [12] by Jung and colleagues showed that of 1076 patients who underwent arthroscopic rotator cuff repair, 483 were treated with screw-in-type bioabsorbable or biocomposite anchors, and 593 were treated with soft anchors. In the screw-in-type anchor group, 16 patients (3.3%, 16/483) experienced anchor pullout during surgery. Of the 593 patients treated by soft anchor insertion, 32 (5.4%, 32/593) experienced anchor pullout. These rates are not significantly different. Intraoperative anchor pullout was much more likely to happen in patients with larger rotator cuff tears, women, older people, or those who had shoulder stiffness before surgery.

#### 3.1.3. Anchor Pullout Has a Relatively Low Incidence

Cummins et al. [8] found three mechanisms by which rotator cuff repairs fail mechanically at the time of revision surgery: rotator cuff suture pullout from the repaired tendon (86.3% of cases), new tears in a different place (9.1% of cases), and complete anchor displacement, or pullout, from the bone (4.5% of cases). 

### 3.2. Risk Factors for Suture Anchor Pullout

The reason why the suture anchor pulls out is the poor stability of anchor fixation in arthroscopic rotator cuff repair. The stability is determined by bone quality, insertion depth, insertion angle, anchor design, the materials used to produce anchors [13,14,15,16,17,18,19] and so on.

#### 3.2.1. Bone Quality

Pullout strength depends on bone mineral density [4,20,21].

Suture anchors have better pullout characteristics when placed in areas of higher bone mineral density (BMD) [4,20,22,23]. However, the use of anchors in patients who are elderly and who may be osteoporotic [24] can potentially increase the likelihood of anchor pullout. The bone quality of the greater tuberosity is one of the factors affecting repair integrity [6]. In patients with poor bone quality, the failure rate after rotator cuff repair is as high as 68% [4,25]. The quality of the proximal humerus bone also deteriorates with age and is more pronounced in patients who have RCT. Djurasovic et al. [7] reviewed 80 cases of failed rotator cuff repair and showed that 10% of them had anchor migration or loosening. Anchor migration is a state between anchor loosening and pullout, and it is incomplete anchor pullout. From these results, we can see that the lower the bone mineral density, the more easily the anchor will cut out of the humerus [4]. These studies provide a theoretical basis for various augmentation technologies.

#### 3.2.2. Anchor Material and Design

The mechanical fixation (pullout strength) of suture anchors is determined by their design, such as the pitch and number of threads, length, size, and overall shape [4,10,26]. Anchors of various designs, materials, and sizes have been invented. The pullout strength can differ according to the material and design of suture anchors [27,28,29].

Anchors made of different materials have different incidences of anchor pullout. Tingart et al. [4] found that the pullout strength of metal screw-type anchors is higher than that of biodegradable hook-type anchors. In addition, most studies using radiotransparent (RT) anchors have reported complications caused by bioabsorbable anchors resulting in bone lysis, defects, and sometimes fractures [30,31,32], which may lead to late pullout. However, osteolysis has no effect on clinical outcomes [33]. Polyetheretherketone (PEEK) anchors, although non-absorbable, also enlarge the peripheral bones significantly more laterally than medially in double-row (DR) repairs [34], which may be the reason why they pull out. In short, the use of anchors of different materials will produce different pullout strengths, which is the reason why anchor pullout occurs under specific circumstances.

A study by Chae and colleagues [35] indicated that high pullout strength was primarily attributed to geometric design factors of the suture anchors, such as greater contact surface area between the anchor threads and surrounding bone, overall length, number of threads, and height of the thread. It is possible that the contact surface area between the anchor threads and surrounding bone is related to other geometric design factors of the suture anchor, such as the overall length, diameter, number of threads, height of the thread, pitch, and helix angle. Chae et al., found that the number and height of threads were positively correlated with the pullout strength among suture anchors of several geometric designs. In fact, the number and height of threads are the most important geometric design factors for increasing the contact surface area between suture anchors and surrounding bone. Their results support the points of view that greater thread-to-bone surface contact leads to greater pullout strength and that screw threads impart improved holding strength due to the increased through contact with the surface of the bone [36,37,38]. Kang et al. [39] reported that a micropore bioabsorbable suture anchor had higher pullout strength, which may have been related to the bone growth induced by the micropore bioabsorbable anchor. Even though more clinical trials need to be conducted to confirm the above assumptions, there is no doubt that these research results point us in a new direction when it comes to anchor design.

Therefore, the clinical application of anchors with different designs will often bring about different pullout strengths and lead to different incidences of anchor pullout. Anchor design determines the stability of anchor fixation, which is one of the reasons why anchor design affects the incidence of anchor pullout.

#### 3.2.3. Number of Anchors (Distance between Anchors)

The relationships of the pullout strength to the anchor material, anchor design, insertion angle, insertion depth, and bone mineral density have been investigated [3,15,16,20,40,41,42]. However, these studies only focused on the pullout strength of one anchor. One study investigated the pullout strength of two anchors instead of one [43], finding that the pullout strength of two anchors was higher.

Kawakami and colleagues [44] showed that in polyurethane and porcine models, the minimum distance between anchors to not reduce the pullout strength was 6 mm, which was less than the previously determined 10-mm separation, and this result was not affected by the different bone qualities, even when applied to osteoporotic bone. When two anchors are placed 4 mm apart, there are two possible reasons for the decrease in pullout strength. First, when two anchors are very close, the cancellous bone around the anchors is not strong enough to support both. Second, in fact, a 4-mm distance means that adjacent anchors will overlap. The contact area between the anchor thread and the cancellous bone decreases as the amount of overlap increases. The contact area of the anchor thread is closely linked to the pullout strength [37,45]. When calculating the distance between two anchors, we mean from center to center, so the minimum distance without decreasing the pullout strength may be different for suture anchors with different diameters. However, due to financial constraints, only two types of suture anchors were examined in this study, compared to the ideal situation of testing all commercially available anchors.

All-suture anchors are biomechanically inferior to screw-in-type anchors [46]. However, Ntalos et al. [47] reported that all-suture anchors and traditional anchors had similar average pullout strengths in an unlimited cyclic model. Moreover, compared with traditional anchors, all-suture anchors have a smaller volume, which allows more of them to be implanted in the same volume of bone [48]. The overall biomechanical performance is improved by sharing the load at multiple fixation points. However, the minimum distance between all-suture anchors seems not to have been reported.

#### 3.2.4. Insertion Angle

Accidental anchor pullout is a common mechanism of repair failure, and its occurrence is affected by bone quality and the implantation technique [49,50]. The relationship between anchor pullout and anchor insertion angle has also been studied. It was widely accepted and understood that placing the anchor at 45° to the insertion surface would display the strongest pullout strength [4,15,18,19,20,51] after Burkhart’s proposal of the deadman theory in 1995 [52].

In 2009, Strauss et al., used cadaveric shoulders to study the effects of anchor insertion angle and rotator cuff tendon repair [49]. The torn supraspinatus tendons were repaired by single suture anchor with an insertion angle of 45° or 90°. The results showed that the rotator cuff repair with the anchor inserted at 90° to the bone surface was stronger than the repair with the anchor inserted at 45°. However, compared with the whole repaired construct, the effect of insertion angle on just anchor pullout strength was of more interest to the researchers.

In 2014, Clevenger et al., tested the pullout strength of anchors with insertion angles from 45° to 135° in 15° increments [14]. According to the findings, anchors set at an acute angle to the pulling axis were substantially weaker than those positioned at an obtuse angle. It did not appear to necessarily match the clinical settings, though, as just one type of synthetic cancellous bone was used and no cortical bone. In 2016, Nagamoto et al., conducted a biomechanical test of anchor insertion angle using the greater tuberosity of porcine humeri and three different densities of synthetic cancellous bones with a 2-mm-thick cortical bone connected to one side. Their findings showed that regardless of bone density, the pullout strength of the anchors implanted at 90° to the bone surface was higher than the anchors inserted at 45° [16].

In the same year, Itoi et al., comprehensively evaluated their laboratory data against previous data and concluded that insertion angles of 45° and 90° were the strongest for threadless and threaded anchors, respectively [53,54]. So, whether threaded or threadless anchors are used should also affect the choice of insertion angle. Threadless anchors provide less friction. In this case, inserting an anchor at 45° had a higher pullout strength than inserting an anchor at 90° or more. In contrast, threaded anchors can provide substantial friction. Therefore, the maximum pullout strength can be obtained by inserting the anchor at 90°.

In 2018, Ntalos et al.’s [47] biomechanical study demonstrated that the maximal force in all-suture and traditional anchors could be detected at a 90° insertion angle. Regardless of the kind, the pullout strength was decreased when they were inserted at more acute (45°) or obtuse (110°) angles. Those differences were not statistically significant, though. They thought that the angle at which the anchor was inserted was not as important in the clinic as people had thought [47].

#### 3.2.5. Size of Rotator Cuff Tear

The retrospective cohort study by Benson and colleagues provided conclusive evidence that patients with larger rotator cuff tears have a significantly higher incidence of anchor pullout. They found [10] that among 251 patients who used metallic suture anchor for rotator cuff repair, six had early anchor pullout, with an overall incidence of about 2.4%. The incidence of rotator cuff tears less than or equal to 3 cm was 0.5%, and the incidence in tears greater than 3 cm was 11%. In large tears, the suture anchor bears higher tension, so the incidence of anchor pullout will also be higher.

#### 3.2.6. Insertion Depth

In a biomechanical study, Bynum et al. [15] showed that changing the insertion depth of the suture anchor affected the mechanical properties and the failure modes of suture anchor constructs. Suture anchors inserted with the suture eyelet deep had premature failure because of construct elongation.

Kirchhof et al. [55] reported that screwing the anchor deeper did not increase the pullout strength. This is because the deep bone mineral density of the greater tuberosity is relatively low. For patients with osteoporosis, this is of no help. Osteoporosis usually involves the patient’s cancellous bone first, resulting in a decrease in cancellous bone quality, followed by cortical bone. Therefore, the deep bone mineral density of the greater tuberosity for patients with osteoporosis is relatively low, and screwing the anchor deeper cannot improve the pullout strength.

Therefore, there is no consistent conclusion on whether increasing the insertion depth of the anchor improves the pullout strength.

#### 3.2.7. The Effect of Corticosteroid Injections on Anchor Pullout Strength

Because RCT patients usually have obvious pain symptoms, corticosteroid injections (CSIs) into the subacromial space have been an important treatment for RCT patients. Puzzitiello et al. [56] showed that for patients who had received CSIs within two weeks, their suture anchor pullout strength decreased significantly after arthroscopic rotator cuff repair. There was no significant decrease after 3 or 4 weeks. These findings suggest that for patients who have received CSIs before surgery, we should ensure that they receive surgery after a certain interval of time.

### 3.3. Anchor Pullout Management

As mentioned above, rotator cuff repair has a high retear rate, and the risk of failure increases with the age of the patient [57] and with the size of the tear [58]. The quality of the proximal humerus also deteriorates with age, and this phenomenon is more common in RCT patients. With the continuous development and popularity of arthroscopic rotator cuff repair, the practice of open rotator cuff repair in the new generation of surgeons is becoming rarer. This section focuses on various management techniques and biomechanical principles for anchor pullout during arthroscopic rotator cuff repair.

#### 3.3.1. Changing the Implant Site of Anchors

For arthroscopic rotator cuff repair, the suture anchor is implanted in the proximal humerus, usually into the greater tuberosity. Many studies have analyzed the bone quality distribution of the greater tuberosity.

In 2003, an in vitro biomechanical study by Tingart et al. [4] demonstrated that, within the proximal part of the greater tuberosity, trabecular bone mineral density of the posterior region and cortical bone mineral density of the middle region were highest, respectively. However, loads to failure in the anterior and middle regions were, on average, 62% higher than the load to failure in the posterior region. They came to the conclusion that cortical bone mineral density was a stronger predictor of pullout strength in the proximal region of the tuberosity than trabecular bone mineral density. The pullout strength might be improved by placing suture anchors in the proximal-anterior and proximal-middle regions of greater tuberosity [20].

Kirchhof et al. [5] performed high-resolution peripheral quantitative CT scanning on 36 cadaver specimens, finding that the volume of highest bone quality was found at the posteromedial aspect. Sakamoto et al. [59] used multidetector row computed tomography to successfully perform an in vivo evaluation of the bone microstructure of the humeral greater tuberosity in patients with rotator cuff tears. They also obtained the same results as Kirchhof et al. According to the findings of both studies, the posterior medial region of the greater tuberosity was the best location for anchor insertion in terms of bone quality. This contradicted the results of Tingart et al. [4]. 

#### 3.3.2. Cement Augmentation for Suture Anchors

Bone grafting or using bone cement to fill the void caused by osteoporotic bone resorption or large cystic changes within the subchondral plate can effectively improve the bone quality of patients undergoing arthroscopic rotator cuff repair. It is very difficult to perform structural bone grafting under arthroscopy, and the pullout strength will not be improved immediately, so it is clinically more feasible to inject bone cement to enhance bone quality and improve pullout strength.

Oshtory and colleagues [60] reported that the pullout strength of suture anchors injected with tricalcium phosphate cement increased by 29%. Giori and colleagues [61] reported a 71% gain in pullout strength with anchor augmentation by polymethyl methacrylate (PMMA) cement. Although the pullout strength was improved, PMMA cement is not bioabsorbable, which may make revision surgery harder. Moreover, PMMA cement produces a thermal effect during the curing process, which may also cause bone necrosis, making the pullout strength uncontrollable in specific cases.

Postl and colleagues [62] reported that the pullout strength of suture anchors injected with bioabsorbable and fiber-reinforced calcium phosphate cement increased by 66.8%. This fiber-reinforced calcium phosphate cement can reach a pullout strength similar to that of PMMA cement but also retains the properties of calcium phosphate cement; that is, it does not produce a thermal effect and is bioabsorbable [63]. This new bone cement combines the advantages of calcium phosphate cement and PMMA cement and is a promising reinforcing material. To be applied in the clinic, it needs to be evaluated in further in vivo experiments.

The biomechanical results above show that it is theoretically tenable to improve the pullout strength of different materials by bone cement augmentation (Table 1).

In fact, not only the material of the bone cement but also the injection method of bone cement has a great impact on the final biomechanical results. Braunstein et al. [64] drilled a hole first, then injected bone cement, and finally implanted an anchor. However, this method can easily lead to the extrusion of bone cement, which is not feasible in an arthroscopic setting. Aziz and colleagues [65] introduced a new bone cement injection method that used an open architecture-type anchor. This method allowed the operator to implant the anchor first and then directly inject bone cement through a cannulated in situ suture anchor with fenestrations. This anchor can make the bone cement interlace and bond with the surrounding bone better, increasing the surface area in contact with the bone, so it may have higher pullout strength. At the same time, we can limit the bone cement injection to the distal end of the anchor, which can effectively reduce the occurrence of bone cement extrusion and the thermal effect on the healing surface, thereby reducing the incidence of bone necrosis. In this way, the method can help to retain the bone quality and improve the pullout strength.

The experimental results of in vitro biomechanical studies also confirm the hypothesis above. Aziz and colleagues [65] reported that the pullout strength increased by 167% when bone cement was injected through an open architecture-type anchor, which was much higher than the pullout strength obtained by using the injection method of Braunstein et al. [64], which was only 45% to 47%, depending on the anatomic location. 

#### 3.3.3. Using All-Suture Anchors

The pullout strength of an all-suture anchor mainly depends on the thickness of cortical bone [66]. Therefore, preoperative cortical bone thickness evaluation and no de-cortication during operation are particularly important to improve the pullout strength of all-suture anchors.

There is controversy about comparisons of pullout strength between all-suture anchors and traditional anchors. Negra and colleagues found that the failure load of all-suture anchors is less than that of traditional anchors, and they also have a significantly greater rate of anchor pullout by various failure mechanisms than traditional anchors [46]. However, this conclusion still needs to be further verified in a representative repair model. On the contrary, Ntalos et al. [47] confirmed that all-suture anchors and conventional anchors have no significant difference in biomechanical effects, and their pullout strength is also similar. 

All-suture anchors have a much smaller volume than the traditional screw-in anchors, which allows us to enhance the pullout strength of the repaired construct by implanting more all-suture anchors in the limited bone [67].

#### 3.3.4. Increasing the Number of Suture Limbs

Shi et al. [68] found that when controlling for the number of sutures, using more suture limbs will result in a higher ultimate failure load. Conversely, when controlling for the number of sutured limbs, they found no significant differences among SR anchored, DR anchored, TOE, and transosseous repairs. In fact, they found that the number of sutures, the number of suture limbs, and the number of mattress stitches were more important in determining the overall strength than the suture structure.

#### 3.3.5. Buddy Anchor Technique

As we know, in patients with osteoporosis, the inserted suture anchors are likely to be unstable. Thus, Brady and Burkhart [36] introduced the buddy anchor technique as a salvage technique: a second anchor is inserted adjacent to the loose anchor to create an interference fit and subsequent higher pullout strength [36,38]. As reported by Denard and Burkhart, the essential mechanism of the buddy anchor system is reinforcement of the pull-out strength by interference fit [69].

One biomechanical study by Horoz et al., supported this technique [38]. They found that in osteoporotic bone, two interlocking suture anchors were stronger than a single anchor. The pullout strength was increased by interlocking a second suture anchor with the first. However, another study contradicted this finding [44]. The opposing view was that placing the two anchors to overlap would reduce the anchor bone contact area and thus reduce the pullout strength [44]. However, the effectiveness of the buddy anchor technique was for the original loose anchor, and the study did not evaluate whether the use of the buddy anchor technique helped to enhance the fixation of the original loose anchor. These two studies were in vitro biomechanical, and more in vivo clinical studies are needed to demonstrate the effect of buddy screwing in the future.

Jung et al. [12] used the buddy screwing technique to augment repair in 16 patients who experienced intraoperative anchor pullout. Three patients had early postoperative failure after buddy screwing. They thought that placing another anchor in an enlarged area tended to result in instability. However, this study was not strictly a randomized controlled trial. The number of cases was also limited since intraoperative anchor pullout was uncommon.

#### 3.3.6. Steinmann Pin Anchoring

Jung et al., invented a new anchor pullout management technique, bar anchoring with a threaded Steinmann pin (BASP) [12]. Using a threaded Steinmann pin (S-pin) (2.3 mm) and sutures, BASP was used to anchor pullouts during surgery. A threaded S-pin was trimmed to a length of 25–30 mm, and the center two-fifths of the S-pin were wrapped with three strands of a No. 2 high-strength suture and tied. A grasper was used to move the S-pin to the pullout site after its short part had been inserted through the anchor insertion portal. A specially made impactor was then attached to the end of the S-pin while being held below the cortical bone of the GT. The suture strands were then withdrawn to cause the S-pin to flip into the cancellous bone of the GT. Firm tension was gradually applied to the strands while observing the S-pin through the GT hole to ensure fixation. The three strands were attached to the S-pin using the Revo knot, a non-sliding knot, and were then used to repair the ruptured tendon.

In this study, the success rate of pullout management was 100% (13/13) for the BASP technique. At 6 months postoperatively, the tendon healing rate in patients undergoing BASP was 92.3% (12/13). 

We can say that the BASP technique achieves satisfactory results both in terms of preventing suture anchor re-pullout and improving the tendon healing rate.

#### 3.3.7. Using an Arthroscopic Transosseous Knotless Anchor

For arthroscopic rotator cuff repair, the most commonly used technology is TOE repair technology. It is not a real transosseous repair technology, and suture anchors are still needed, which means that this technology still brings a risk of anchor pullout, especially in elderly patients and patients with osteoporosis.

Therefore, some surgeons have developed an arthroscopic transosseous knotless (ATOK) anchor to realize true transosseous repair through arthroscopic technology. A noninferiority trial by Sandow and colleagues [70] showed that none of the 15 patients who received the ATOK anchor for rotator cuff repair had anchor displacement or anchor pullout. 

Compared with the widely used TOE repair technology, ATOK anchor repair can potentially reduce the incidence of anchor pullout. The effect of this technology also needs to be validated by randomized controlled trials with larger sample sizes.

#### 3.3.8. Transosseous Suture Repair Technology

The advent of suture anchors helped popularize arthroscopic rotator cuff repair due to the ease and speed of operation and their facilitation of instrumentation [20]. However, arthroscopic rotator cuff repair with suture anchors is not reliable for patients with osteoporosis, so some scholars have readopted transosseous suture repair. Of course, the current transosseous suture repair is not the same as the earlier arthrotomy but is performed under arthroscopy.

Randelli et al. [71] conducted a randomized controlled trial to compare the effectiveness of arthroscopic transosseous repair to single-row suture anchor repair. The two procedures produced equal results in terms of functional and radiological outcomes. Moreover, transosseous repair was found to reduce pain more quickly in the first month after surgery. 

A matched cohort study by Srikumaran et al. [72] showed that in terms of patient-reported results, shoulder range of motion, and structural integrity, there are no differences between transosseous and transosseous equivalent suture-bridge rotator cuff repair procedures. The operating time was the same for all procedures. However, future randomized controlled trials are still needed to further demonstrate the equivalence of the two techniques.

These results demonstrate that arthroscopic transosseous repair can achieve the same results as arthroscopic rotator cuff repair with suture anchor in all aspects, and as a repair technique using only sutures without anchors, it can be used as an alternative treatment option for patients with anchor pullout. However, in patients with osteoporosis, suture cutting of the bone may also lead to failure of the repair.

In order to avoid bone cutting, some authors advise using a broader suture, such as a 2 mm tape, rather than the thinner No2 wire [73]. Due to its ideal viscoelastic properties [74] and broader contact surface with bone and soft tissue, this tape would exert the same force but less pressure at the contact region. 

A cohort study by Beauchamp et al. [75] showed that arthroscopic transosseous repair using 2 mm tape material achieved significant mid-term functional improvement in this group of patients, with results statistically unaffected by larger tear size (>3 cm) or older age (≥65 years), which also happen to be risk factors for anchor pullout after rotator cuff repair with suture anchor. Therefore, arthroscopic transosseous repair using 2 mm braided suture tape could be an alternative surgical option to reduce the risk of anchor pullout for these two types of patients.

## 4. Discussion and Clinical Inspirations

This review organizes the body of knowledge on anchor pullout through a literature review. However, there are still some issues that need to be addressed in this area.

There are very few studies on the incidence of anchor pullout. Although the overall incidence of anchor pullout is not too high [8,9,10,11,12], studies of pullout rates under certain conditions, such as in patients with poor bone quality, needed, which are important for clinicians’ preoperative decision making.

Additionally, we present the risk factors for anchor pullout. The effect of bone quality on pullout strength is relatively well established, and clinical studies with larger sample sizes would provide stronger support for the existing view. For anchor material and design, anchor insertion angle and depth, the existing findings are mainly from in vitro biomechanical studies [30,31,32,33,34,35,36,37,38,39,40,46,47,48], which do not fully simulate the clinical situation, and more in vivo clinical studies are needed to confirm the existing findings in the future. The minimum distance (center-to-center) of suture anchors without decreasing the pullout strength varies with anchor diameter [44]. The available studies only tested two types of suture anchors rather than all commercially available suture anchors. The minimum distance between all suture anchors has not been reported yet. Since the level of evidence from existing studies is low, future randomized controlled trials are needed to evaluate the effect of rotator cuff tear size and CSIs on pullout strength [10,56]. 

The focus of this review is corresponding management for suture anchor pullout. For the anchor implantation site, Kirchhof and Sakamoto only evaluated the distribution of total bone mineral density in the greater tuberosity without performing the corresponding biomechanical tests, and their findings cannot be used as predictors of anchor pullout strength [5,59]. From this point of view, the conclusions of Tingart et al. [4] seem to be more credible. Their study showed that placing the anchor in the anterior and middle regions proximal to the GT resulted in an average load to failure 62% higher than placing it in the posterior region [20]. Prospective clinical trials are necessary to understand whether the available managements can reduce anchor pullout rates and improve patient prognosis. By comparing the in vitro biomechanical data, we found that PMMA bone cement provides greater pullout strength compared to various new bone cements, despite its various drawbacks [60,62,63]. For the bone cement injection method, the injection of bone cement through an open architecture-type anchor is also superior to the traditional method of drilling a hole first, then injecting bone cement, and finally implanting the anchor [64,65]. However, the arthroscopic application of this technique still needs to overcome some technical difficulties, which require additional in vivo clinical studies. When a patient is at high risk for anchor pullout, we can assess their cortical bone thickness preoperatively, and if the cortical bone quality is good, we can use all suture anchors for rotator cuff repair because they are small and can be implanted in greater numbers, which can improve the overall pullout strength [66,67]. Buddy screwing, BASP, and ATOK are three relatively new techniques. They are not only theoretically valid but have also been demonstrated in several studies. In fact, buddy anchor technique is a controversial technique. In the case of a small sample size, its in vivo application has a failure rate of 19% (3/16) [12], but it remains one of the few means of remedy in the event of intraoperative anchor pullout. In contrast, better pullout strength was achieved using the BASP technique and the ATOK anchor, and neither in vivo study reported anchor displacement or pullout [12,70]. However, these two techniques are more complex to perform, and in practice, buddy screwing remains a trusted and relatively simple remedy. However, neither of these studies were strict randomized controlled trials, nor was the sample size large enough [1,36,38,44,69,70]. Relatively speaking, arthroscopic transosseous suture repair is a more established technique. Its equivalence to suture anchor repair was also confirmed by a randomized controlled trial [71]. Arthroscopic transosseous suture repair does not involve the use of suture anchors at all, which is very suitable for patients at high risk of anchor pullout. However, this technique also presents a new problem; that is, the sutures may cut the osteoporotic bone, leading to repair failure. However, there are no studies on the probability of anchor pullout and bone cutting with suture anchor repair and transosseous suture repair for the same bone quality, respectively. Some cohort studies suggest that the use of wider sutures may reduce the risk of transosseous suture repair failure [73,74,75]. However, due to the uncertainty about the incidence of bone cutting, in vitro biomechanical studies using a severe osteoporosis model may help to increase positive results and thus help us better evaluate the effectiveness of this approach.

In conclusion, not many of the management techniques have been widely used in clinical practice. Since most are derived from in vitro biomechanical studies, in vivo randomized controlled trials with larger sample sizes are needed to confirm whether they can ultimately benefit patients.

## Figures and Tables

**Figure 1 jcm-11-06870-f001:**
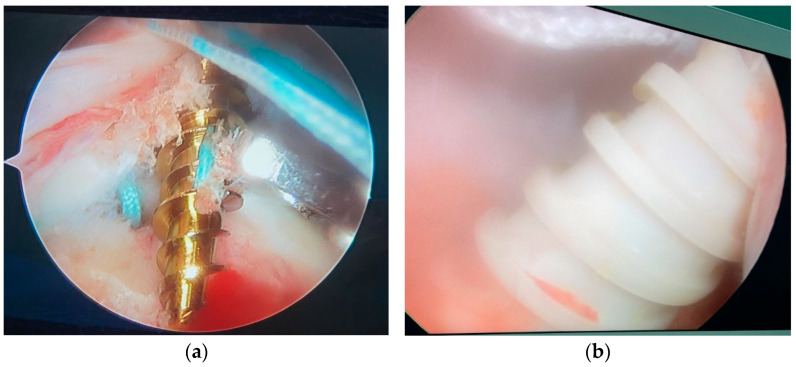
(**a**) Intraoperative metallic suture anchor pullout; (**b**) intraoperative polyetheretherketone suture anchor pullout.

**Table 1 jcm-11-06870-t001:** Pullout strength increment for augmentation with different types of bone cement.

Study	Types of Bone Cement	Testing Model	Anchor Type	Percentage Increase (%)	*p* Value
Giori et al. [61]	PMMA bone cement	Cadaveric humerus	Metal screw-like suture anchors (5-mm Fastin RC; Mitek, Norwood, MA, USA)	71	*p* = 0.02
Oshtory et al. [60]	Bioabsorbable tricalcium phosphate cement	Cadaveric humerus	Metal screw-like suture anchors (5-mm Fastin RC; Mitek, Norwood, MA, USA)	29	*p* = 0.027
Postl et al. [62]	The bio-absorbable and fiber-reinforced calcium phosphate cement	Cadaveric humerus	titanium suture anchors (Corkscrew FT 1 Suture Anchors, Arthrex, Naples, FL, USA)	66.8	*p* < 0.001

## Data Availability

No new data were created or analyzed in this study. Data sharing is not applicable to this article.

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
