# Peer review of "Risk Factors and Corresponding Management for Suture Anchor Pullout during Arthroscopic Rotator Cuff Repair"

_jcm, 2022, doi:10.3390/jcm11226870_

Round 1

Reviewer 1 Report

I congratulate the authors on their attempt at  comprehensive review of anchor pullout and management.  However, there are numerous issues with the manuscript.

-There are many locations with incorrect English grammar or incorrect use of English words.  (lines 38, 40, 49, 63, 66-68, 70, 76, 78-81, 94-95 and many more throughout manuscript).  Would recommend thorough review prior to resubmission.

Key points: this section is very general for "key points" and doesn't actually mention any of the actual key points (i.e. 3 causes of anchor pullout.  what to do with anchor pullout).

Line 94-95 Would eliminate this title as a separate section

Line 204-7 No data is presented here. Would like to see how larger tear leads to failure by anchor pullout and not failure of suture/tissue interface or tendon/bone interface.

Line 222: CSI isn't defined until later. need to write out

Line 238-252.  Is one cadaver study really enough to dictate where anchors should be placed> Anchors should be placed where ideal to fix he tear.  How about using a smaller awl?  Using all suture anchors with ~1mm drill hole?

Line 306-321:  If there is no clinical data that repair construct impacts anchor pullout (the subject of this paper), this section should be eliminated.

Line 372-5: The authors mention a new technique but do not describe anything about it.

Line 390-418: There is no place in this review of literature for describing a novel technique with no data behind it.  This would be better served as a technique paper of this novel technique.

Author Response

1.There are many locations with incorrect English grammar or incorrect use of English words.  (lines 38, 40, 49, 63, 66-68, 70, 76, 78-81, 94-95 and many more throughout manuscript).  Would recommend thorough review prior to resubmission.

Response:We apologize for the poor language of our manuscript. We worked on the manuscript for a long time and the repeated addition and removal of sentences and sections obviously led to poor readability. We have now worked on both language and readability and have also involved native English speakers for language corrections. We really hope that the flow and language level have been substantially improved.

2.Key points: this section is very general for "key points" and doesn't actually mention any of the actual key points (i.e. 3 causes of anchor pullout.  what to do with anchor pullout).

Response:Considering the overall logic of this review, we made significant changes to the structure of this review, adding a discussion section to elaborate on the key points of this review, and the original "key points" section was directly deleted.

3.Line 94-95 Would eliminate this title as a separate section

Response:We have changed the original title to "Anchor pullout has a relatively low incidence"(Line 95).

4.Line 204-7 No data is presented here. Would like to see how larger tear leads to failure by anchor pullout and not failure of suture/tissue interface or tendon/bone interface.

Response:We have added data from the original literature to support our view(Line 225–232). However, it does not mention the effect of larger rotator cuff tears on the suture/tissue interface or tendon/bone interface, and the main purpose of this section is to demonstrate that a large rotator cuff tear may be a risk factor for anchor pullout.

5.Line 222: CSI isn't defined until later. need to write out

Response:It has already been corrected(Line 248).

6.Line 238-252.  Is one cadaver study really enough to dictate where anchors should be placed> Anchors should be placed where ideal to fix he tear.  How about using a smaller awl?  Using all suture anchors with ~1mm drill hole?

Response:We have added a citation to a biomechanical study that introduces a different perspective on the optimal anchor implantation site(Line 269–277). However, none of these three studies is sufficient to dictate where anchors should be placed. We believe that in vivo clinical trials are still needed to determine the anchor implantation site. No relevant studies were retrieved for using a smaller awl, but we believe that this is related to the reduced decortication, and we cite a study by Tingart et al. in this section(Line 269–277). They suggest that better cortical bone quality provides greater pullout strength. As for using all suture anchors with ~1mm drill hole, we also believe that this reduces the damage to the bone. We describe this idea in the section of using all-suture anchors(Line 324–327):preoperative cortical bone thickness evaluation and no decortication during operation are particularly important for improving the pullout strength of all-suture anchors.

7.Line 306-321:  If there is no clinical data that repair construct impacts anchor pullout (the subject of this paper), this section should be eliminated.

Response:We really did not retrieve the relevant clinical data from the PubMed database. So we have removed this argument.

8.Line 372-5: The authors mention a new technique but do not describe anything about it.

Response:By reviewing the original literature, we have added a detailed description of this new technology and presented data to support it(Line 386-403).

9.Line 390-418: There is no place in this review of literature for describing a novel technique with no data behind it. This would be better served as a technique paper of this novel technique.

Response:We have removed the description of this novel technique due to the unavailability of literature reports and related data. We have added the literature on transosseous suture repair for citation and recommend transosseous suture repair as a way to deal with anchor pullout(Line 426-452).

Reviewer 2 Report

The authors present a literature review newly developed techniques that are used to solve anchor pullout in patients with osteoporosis after rotator cuff repair surgeries. Timely reviews help researchers with research. Submitted study is well organised and have a potential to contribute to the literature, however, it requires revision before being a publication.

The authors state that aim is to review newly developed techniques. This is a vague statement. What were the inclusion criteria in the study?

What were the publication years that the authors considered in the review for the papers?

What databases were used to search the literature?

Why do the authors not perform comparative analyses using similar studies and present the results in tables?

One would expect to see future directions and critical review of the literature in a review study. The authors can include possible future directions and critical comments at the end of their review.

Author Response

1.The authors state that aim is to review newly developed techniques. This is a vague statement. What were the inclusion criteria in the study?

Response:The literature search included rotator cuff repair as well as anchor pullout strength including all articles published in PubMed until September 2021. The following search terms were used alone and in combination: "Rotator cuff repair", "anchor pullout", and "pullout strength”.Articles of all in vitro biomechanical and clinical trial levels in English were included. After assessing all abstracts (n = 275), the full text and the bibliographies of the relevant articles were analyzed about the questions posed (n = 80). Articles including outcomes without the area of interest were excluded (n = 22). Specifically, publications from 1990 to 2021 were included. The final literature research revealed 58 relevant articles.

Thanks to your suggestion, we have therefore added a section(Line 48-61) dedicated to the data collection for this review.

2.What were the publication years that the authors considered in the review for the papers?

Response:Publications from 1990 to 2021 were included because of the advancements in biomechanics, surgical treatments, and improved understanding of pullout strength for suture anchors.

Thanks to your suggestion, we have therefore added a section(Line 48-61) dedicated to the data collection for this review.

3.What databases were used to search the literature?

Response:We used the PubMed database to search the literature.

Thanks to your suggestion, we have therefore added a section(Line 48-61) dedicated to the data collection for this review.

4.Why do the authors not perform comparative analyses using similar studies and present the results in tables?

Response:The content of this review covers three aspects of anchor pullout. The types of literature included in each section are also very different. Their study types, inclusion criteria, and interventions were very different, which prevented us from conducting comparative analyses.

Thanks to your suggestion, we have performed a comparative analysis of three studies using different types of bone cement for bone cement augmentation, which is illustrated in the table(Line 336).

5.One would expect to see future directions and critical review of the literature in a review study. The authors can include possible future directions and critical comments at the end of their review.

Response:We have added a discussion section to provide potential future directions and critical comments as a result of your valuable suggestion(Line 454-497). 

Reviewer 3 Report

Thank you for the invitation to review the interesting paper with the title 'Risk Factors and Corresponding Management for Suture Anchor Pullout during Arthroscopic Rotator Cuff Repair'.

The authors address a relative common problem in shoulder arthroscopy, they analyze the etiology and suggest possible solutions for anchor pullout during arthroscopic rotator cuff repair. Although the present paper is a narrative review and no specific guidelines are required, there are some methodological issues that need to be discussed before publication.

Abstract must be revised because most sentences are written twice and its structure does not follow the whole paper.

References are not following the order they appear in the text. For example, the first reference (line 46) is number 22. Many references are not cited anywhere in the text.  

There are many paragraphs based on the existing literature, where a relative reference is missing (i.e. lines 56-61).

The paper is divided into two main parts: 1) risk factors for suture anchor pullout and 2) anchor pullout management. These sections must be revised because there are overlapping sections and although the information they provide is adequate, it is very difficult for the reader to have a clear answer to the issues raised by the authors. For example lines 364-371 are a repetition of lines 183-202.

A more structured form of the paper should be considered by the authors with clear suggestions based on the existing literature.

Author Response

1.Abstract must be revised because most sentences are written twice and its structure does not follow the whole paper.

Response:In line with the logic of this review, we have revised the abstract and removed duplicate contents(Line 9-25).

2.References are not following the order they appear in the text. For example, the first reference (line 46) is number 22. Many references are not cited anywhere in the text.  

Response:Thanks to your valuable suggestion, we have therefore rearranged the order in which references appear and ensured that all references are cited in the text.

3.There are many paragraphs based on the existing literature, where a relative reference is missing (i.e. lines 56-61).

Response:Thanks to your suggestion, we have added references to the literature in the corresponding paragraphs.

4.The paper is divided into two main parts: 1) risk factors for suture anchor pullout and 2) anchor pullout management. These sections must be revised because there are overlapping sections and although the information they provide is adequate, it is very difficult for the reader to have a clear answer to the issues raised by the authors. For example lines 364-371 are a repetition of lines 183-202.

Response:Based on your suggestions and consideration of the logic of this review, we have combined the duplicated sections(Line 120-158,Line 159-185,Line 187-224).

5.A more structured form of the paper should be considered by the authors with clear suggestions based on the existing literature.

Response:We have made major revisions to the content and overall structure of this review, and at the end of this review we give our critical comments about this area, as well as suggestions for future research directions(Line 454-497).

Round 2

Reviewer 1 Report

Authors have made appropriate edits. In abstract authors state there is no data on this subject. I would disagree that there is some but limited data. 

organization of paper much improved. 

Author Response

Authors have made appropriate edits. In abstract authors state there is no data on this subject. I would disagree that there is some but limited data.

organization of paper much improved

Response:Thanks to your suggestion, we have therefore made the corresponding change (Line 21) .

Reviewer 2 Report

The authors respond to the reviewer's comments sufficiently. Submitted study can be accepted as a publication.

Author Response

The authors respond to the reviewer's comments sufficiently. Submitted study can be accepted as a publication.

Response:Thanks to your suggestion, we sincerely hope that we can do better.

Reviewer 3 Report

The authors addressed all the issues raised in the previous round of the review process. The manuscript is now well structured, duplicate and overlapping sections are removed and the order of the references is corrected. The revised manuscript provides valuable information about the risk factors of anchor pullout after arthroscopic rotator cuff repair and emphasizes on its management by summarizing possible solutions that have been published in the literature. I believe the revised version of the manuscript entitled 'Risk Factors and Corresponding Management for Suture Anchor Pullout During Arthroscopic Rotator Cuff Repair' merits publication in Journal of Clinical Medicine. 

Author Response

The authors addressed all the issues raised in the previous round of the review process. The manuscript is now well structured, duplicate and overlapping sections are removed and the order of the references is corrected. The revised manuscript provides valuable information about the risk factors of anchor pullout after arthroscopic rotator cuff repair and emphasizes on its management by summarizing possible solutions that have been published in the literature. I believe the revised version of the manuscript entitled 'Risk Factors and Corresponding Management for Suture Anchor Pullout During Arthroscopic Rotator Cuff Repair' merits publication in Journal of Clinical Medicine.

Response:Thanks to your suggestion, we sincerely hope that we can do better.